# BiGAMi: Bi-Objective Genetic Algorithm Fitness Function for Feature Selection on Microbiome Datasets

**DOI:** 10.3390/mps5030042

**Published:** 2022-05-23

**Authors:** Mike Leske, Francesca Bottacini, Haithem Afli, Bruno G. N. Andrade

**Affiliations:** 1Department of Computer Sciences, Munster Technological University, MTU/ADAPT, T12 P928 Cork, Ireland; mike.leske@gmail.com; 2Department of Biological Sciences, Munster Technological University, MTU, T12 P928 Cork, Ireland; francesca.bottacini@mtu.ie

**Keywords:** feature selection, genetic algorithm, human health, machine learning, microbiome

## Abstract

The relationship between the host and the microbiome, or the assemblage of microorganisms (including bacteria, archaea, fungi, and viruses), has been proven crucial for its health and disease development. The high dimensionality of microbiome datasets has often been addressed as a major difficulty for data analysis, such as the use of machine-learning (ML) and deep-learning (DL) models. Here, we present BiGAMi, a bi-objective genetic algorithm fitness function for feature selection in microbial datasets to train high-performing phenotype classifiers. The proposed fitness function allowed us to build classifiers that outperformed the baseline performance estimated by the original studies by using as few as 0.04% to 2.32% features of the original dataset. In 35 out of 42 performance comparisons between BiGAMi and other feature selection methods evaluated here (sequential forward selection, SelectKBest, and GARS), BiGAMi achieved its results by selecting 6–93% fewer features. This study showed that the application of a bi-objective GA fitness function against microbiome datasets succeeded in selecting small subsets of bacteria whose contribution to understood diseases and the host state was already experimentally proven. Applying this feature selection approach to novel diseases is expected to quickly reveal the microbes most relevant to a specific condition.

## 1. Introduction

The past decade has shown a gradual introduction of classical and advanced machine-learning (ML) methods applied to bioinformatics, enabling the use of feature selection, and regression and classification algorithms in the microbiome field [1]. These methods allowed some of the first successes in identifying key features, including microbiome taxa or gene abundances, and using them to classify or predict environmental conditions based on the microbiota itself [2,3]. ML approaches have also been applied to studies with human populations to investigate the link between the host’s microbiota composition and health conditions, including obesity [4], colorectal cancer [5], liver cirrhosis [6], type 2 diabetes [7], bacterial vaginosis [8], and irritable bowel disease (IBD) [9]. These studies achieved high scores classifying individuals in groups regarding conditions, with either Metabarcoding (16s rRNA coding genes) or deep-sequencing metagenomics data, suggesting the microbiome as a potential source of biomarkers for diagnostics or even treatment using probiotics [10].

Microbiome datasets are small when compared to other fields that benefit from artificial intelligence (AI), with often a few dozen to a few thousand data points due to sampling, experimental, and sequencing costs, which could go as high as hundreds of thousands of dollars to even millions [11,12]. However, microbiome datasets have yet another characteristic that can increase this challenge. They normally have thousands to millions of features to be analyzed, therefore being sparse and highly dimensional [11]. Microbiome features may represent genetic markers grouped into operational taxonomic units (OTUs) or amplicon sequence variants (ASVs) [13,14], together with the corresponding microbial taxonomy classification, or gene abundance and activity levels, gene function, functional pathways, and even protein abundance [3,15,16], having a distinct set of challenges and biases that must be considered in a pre-processing step.

Such data characteristics are likely to cause issues if not properly addressed, a phenomenon known as the “curse of dimensionality” [17], and the high dimensionality of microbiome datasets has often been addressed as a major difficulty in the application of ML algorithms [18]. Feature selection is a commonly used methodology to improve ML algorithm performance in classification and regression by tackling the high dimensionality problem; however, algorithms such as forward selection or backward elimination [19] quickly result in an unmanageable computational complexity due to a large number of microbial features being tested. Other methodologies, such as principal component analysis (PCA), linear discriminant analysis (LDA), and multidimensional scaling, have been applied to the microbiome field [20,21,22]. Collectively, these are known as dimensionality reduction methods, and they reduce the data input by projecting it to a lower dimensional subspace. However, although these methods do reduce the burden of high dimensionality, part of the original information is lost forever.

To address this issue, studies have recently started to implement genetic algorithm (GA) approaches to search for subsets of predictive microbiome features, which in turn lead to an increased model performance. Unlike classical feature selection methods, which focus on sequential addition or removal of features, a GA-based approach has the potential to evaluate feature interactions that are more complex. GA represents evolutionary search methods inspired by Darwinian principles [23,24], and the adoption of GAs is especially popular for finding heuristic solutions to problems that are computationally intractable otherwise (TSP) [25], making GA a potential solution for feature selection on microbiome datasets. This method was successfully applied to search for a subset of vaginal microbiome features to detect bacterial vaginosis using the genetic and evolutionary feature selection (GEFeS) [26] and to select a fixed number of highly predictive features from small, medium, and large-sized omics datasets [27]. Genetic algorithm was also applied after a PCA-based dimensionality reduction with a fixed number of principal components, improving the prediction accuracy [28].

Herein we explore the use of GA for feature selection by addressing the dimensionality problem of microbiome datasets with a bi-objective genetic algorithm to select subsets of microbiome features for classification models. While optimizing for the classification performance of a certain feature subset (objective 1), the fitness score of a potential GA solution is penalized proportionally to the number of the selected features (objective 2). This way, the bi-objective GA search process is actively guided to be optimized for identifying the smallest best performing feature subsets. We also do not restrict the size of the feature subset to a predefined number but allow the search optimization process to grow or shrink the number of selected features according to the selection, crossover, and mutation steps of the GA evolutionary search.

Our method, called “BiGAMi-Bi-Objective Genetic Algorithm Fitness Function for Feature Selection on Microbiome Datasets”, was implemented in Python and released under the open-source MIT license on GitHub (https://github.com/mikeleske/BiGAMi (accessed on 10 April 2022)). The method was tested using four publicly available datasets, transformed in relative abundance, and centered log-ratio (CLR). We compared our results with the baseline scores published elsewhere [26]. The performance of a classical sequential forward selection (SFS) algorithm, a k-best selection based on statistical properties, and GARS, a GA-based feature selection library focusing on optimizing fixed feature subset lengths, were assessed and compared to BiGAMi.

## 2. Materials and Methods

### 2.1. Data Retrieval and Pre-Processing

The microbiome datasets used in this study were retrieved from the Microbiome Learning Repo [29] (https://knights-lab.github.io/MLRepo/ (accessed on 15 March 2021)), a public repository of microbiome data to be used for regression and classification tasks. They were generated by studies aiming to identify associations between microbiome and health status, such as the investigation of the relationship between microbiome, colorectal cancer, and cirrhosis [6,30], and to investigate differences in the vaginal microbiota of human populations [31]. The datasets included three microbiome matrices containing a range of 586-8483 operational taxonomic units (OTU) and their counts generated by mapping against two different reference databases, the Greengenes 97 (GG97) [32] and RefSeq [33]. The repository also made available the corresponding sample metadata for each dataset, which contained crucial information for the classification procedures, such as the patients’ health status. Table 1 provides a summary of the datasets and classification tasks (I-IV) used to evaluate the performance of our BiGAMi approach.

The matrices were transformed into relative abundance (percentage) and using a compositional data analysis (CoDA) method called centered log-ratio (CLR), through custom Python scripts and the Scikit-bio package (https://github.com/biocore/scikit-bio (accessed on 24 April 2021)). The datasets were scaled using the Scikit-learn MinMax function [34], followed by the selection of the 128 most important microbiome features using the SelectKBest function. Each dataset was then split into 6 parts of equal sizes (6-fold), where the first 5 were used as the training set, and the last was used as a hold-out test set and used to perform classification experiments for each task (Figure 1).

### 2.2. Bi-Objective Genetic Algorithm Fitness Function and Implementation (BiGAMi)

GA represents evolutionary search methods inspired by Darwinian principles. Each GA starts with the creation of an initial population of a predefined number of individuals, which are generated in the form of chromosomes and represent potential solutions to the search problem. The GA search process iterates for a predefined number of generations, or until a stopping criterion is met, through the following steps: (a) Each individual in the GA population is evaluated on how well its chromosome solves the computational problem (Fitness Evaluation); (b) Chromosome pairs are selected for mating based on the individuals’ fitness ranking (Selection); (c) Mating pairs partly exchange genes from their chromosomes to generate offsprings for the next generation (Crossover); (d) new offsprings experience random mutations in one or more genes to introduce novelty into the population (Mutation) (Figure 2).

To perform such a feature selection, we first have to encode the GA individuals’ chromosomes as a binary string of genes, included or not in the fitness evaluation. A gene is set to the value **1** if the associated feature is included in the fitness evaluation process and set to **0** otherwise. We used a sparse chromosome initialization strategy for the initial GA generation that activates only a small fraction of the dataset features per chromosome. This was necessary, since a random initialization of the chromosome could result in unnecessarily large feature subsets and the activation of mostly irrelevant features.

For each chromosome, a separate *Scikit-learn* stochastic gradient descent classifier (SGDClassifier) was trained to predict the target class, e.g., phenotype or disease state, using k-fold cross-validation to evaluate the chromosome’s overall fitness score. The chromosome-specific classifier’s feature coefficients table was then used to reset to **0** the chromosomes containing genes with no significant relevance to the classification task. The crossover operation used the one-point mating option to minimize the risk of producing exact copies of the mating individuals due to their chromosome sparsity. In addition, a custom mutation function was implemented to provide equal chances to either switch off an active gene or switch on an inactive gene. Lastly, a bi-objective fitness function, which considered the classification metric and penalized the usage of larger feature subsets, was used to evaluate chromosome fitness. The bi-objective fitness function was implemented using the following formulae:fitness score = x × metric + y × (selected features/total features) (1)
metric = avg(cv_score) + min(cv_score) (2)

In the above formula, *metric* represents the sum of the average and minimum k-fold cross-validation AUC scores, while *features* are relative abundance or CLR values, and x = 1 and y = −1. For each input data, we executed 25 GA runs with a population size of 300 for 10 generations. The 6-fold data split remained consistent across its generations for each separate GA run. In each generation, the individuals’ performances were evaluated by a 5-fold cross-validation (CV) using a SGDClassifier on the training data (folds 1 to 5).

The sum of the 5-fold average and minimum scores was reported as this individual’s metric, which provides a larger optimization (maximization) space compared to relying on the k-fold average alone. Each SGDClassifier used the “log” loss function, the L1 penalty, and was restricted to 500 iterations. After each generation, the best performing individual (according to 5-fold CV using the bi-objective fitness function) was evaluated against the hold-out test set and promoted into the next generation, a strategy known as the elitism concept. After each GA search, the individual with the highest bi-objective evaluation metric on the test set was identified, resulting in 25 high-performance individuals per input data.

GA individual selection for crossover was based on a tournament selection method with size 3. The crossover operation was set to one point to reduce the risk of selecting only patches of inactive genes due to chromosome sparsity. The probability of 2 selected individuals producing offspring was set to 0.8, a common default value for the crossover probability in GA frameworks. Likewise, the probability of an offspring undergoing a mutation was set to 0.8. The sparse gene activation per chromosome would have given default mutation operators a bias toward activating inactive genes. Therefore, a custom mutation operator was implemented to ensure that after each successful crossover, a mutation operation with a 50% chance to activate or deactivate a single random gene is executed on the new chromosome. The Python package *DEAP* [35] was used as the core framework for the genetic algorithm and evolutionary search process. Table 2 lists the essential parameters together with their respective values to initialize and execute the GA runs.

### 2.3. Other Feature Selection Implementations

In order to test the efficiency of BiGAMi, we compared it against 3 other feature selection methods: a classical sequential forward selection (SFS) [36] implementation, the *Scikit-learn* SelectKBest methodology, and GARS, a different feature selection method that applies a GA-based approach.

SFS is a well-known feature selection methodology and represents a simple and greedy feature selection algorithm where, in each iteration of the process, the single feature that improves the target metric the most is added to the list of the selected features until a stopping criterion is met, e.g., the maximum feature subset size is reached. SFS was applied using the *MLxtend* Python library [36], using the same input data used by BiGAMi and 5-fold cross-validation (based on ⅚ of the dataset size). A total of 25 SFS runs were executed per input to identify, stepwise, the 1 to 32 most important features using the training dataset and subsequently evaluate the test set. For each SFS run, the best performing feature subset was reported, including the test set performance and the selected features.

The *Scikit-learn* SelectKBest methodology with a Chi2 scoring function was applied to extract the *k* = [4, 8, 12, 16, 20, 24, 28, 32] most important features per data input. As this feature selection approach is purely based on statistical properties of the underlying data, multiple runs with the same value for *k* result in the same feature subset. Therefore, for each value of *k*, 25 SGD classifiers were trained on different training data splits, and the average performance of the 25 respective tests is associated with the matching value of *k*.

GARS is a GA-based feature selection framework implemented in R that works with a fixed chromosome length of selected features, i.e., any given GARS run specifies how many features should be evaluated by each GA individual. The reference implementation for the GARS publication provides an individual feature subset per cross-validation fold (https://github.com/BioinfoMonzino/GARS_paper_Code (accessed on 8 May 2022). All classification tasks were performed using the scikit-learn SGDClassifier with L1 penalty.

## 3. Results

### 3.1. Feature Selection Using BiGAMi

The use of BiGAMi to reanalyze consolidated data allowed us to identify and select small subsets of highly informative microbiome features (OTUs), which, when used for classification tasks, greatly improved most classification scores, or at least obtained the same scores of the original studies with considerably fewer features. The results listed here represent the average performance and OTU subset sizes of the best performing solutions found across the 25 BiGAMi runs per data input.

For the Kostic colorectal cancer dataset (Task I), BiGAMi was able to reduce the number of OTUs from 3228 to 12–18 (GG97) and from 908 to 8–17 (RefSeq) while increasing the AUC score from 0.74 to 0.93–0.95 and from 0.69 to 0.84–0.86, respectively, a significant increase in classification power while reducing the number of features by 50–200×. For the vaginal Nugent category dataset (Task II), the baseline AUC score was already 0.99; however, we achieved similar scores of 0.99 using 8–9/1083 (GG97) and 0.98 using 6–7/586 (RefSeq) OTUs, on average. The same behavior was observed when using the same dataset to classify the host’s ethnicity in black or white groups (Task III), an increase of 0.64 to 0.73–0.77 (GG97) and from 0.70 to 0.73–0.75 (RefSeq) while using 11–16/1083 and 10–14/586 OTUs, on average, and when classifying healthy vs. patients with cirrhosis (Task IV), increasing the AUC score from 0.92 to 0.93–0.94 while using 4–12/8483 OTUs, a reduction of 706×, on average (Table 3, Figure 3 and Figure 4). For each classification task, a single data input, indicated by an * symbol, was selected for further analysis, according to the bi-objectivity feature selection approach of BiGAMi. For task III, preference was given to the RefSeq + CLR data input due to the species-level resolution of the RefSeq datasets.

When compared to SFS, BiGAMi achieved a marginally superior or equal classification score for 12 out of 14 experiments, the only exception being the RefSeq annotated dataset for task I. In comparison with the SelectKBest results, BiGAMi shows an improved classification score for 9 out of 14 experiments. Where SelectKBest marginally outperformed BiGAMi by 0.01–0.02 AUC classification metric, SelectKBest only achieved this by selecting substantially larger feature sets of 1.2× to 2.4× of the feature subset sizes BiGAMi selected.

BiGAMi also outperformed GARS. While GARS successfully reduced the number of OTUs, identifying mean subsets of 64.7, 44.5, 46.7, and 56 OTUs across the four experiments, it achieved low classification scores when they were used in the GARS classification model (random forest) (mean of 0.60, 0.89, 0.56, and 0.81 for tasks I-IV, respectively) (Table 3). BiGAMi used, on average, 21% of the OTU used by GARS across the four experiments, while achieving a high classification score (mean of 0.89, 0.98, 0.74, and 0.93 for tasks I–IV, respectively). When executed against the mid-size GARS dataset, which includes approximately 700 features and thus meets well the dimensions of microbiome datasets, BiGAMi achieved an average performance of 0.91 AUC with an average of 6 selected features across 25 GA runs, whereas GARS achieved a classification performance of 0.81 AUC using 9 features.

Figure 5 summarizes BiGAMi performance in comparison to the other feature selection approaches.

### 3.2. Taxonomy Annotation of Feature Subsets

The taxonomic annotation up to the species level was further used to identify and explore the microorganisms selected as important for classification models applied to tasks I–IV (Appendix A) (Table 3, data inputs marked with *). Microorganisms that appeared in fewer than five of a task’s best performing GA individuals were excluded. For task I, where the classification performance based on the Greengenes 97 dataset outperformed the RefSeq dataset, the translation of OTU IDs to taxonomic annotation often resulted in the identification up to the family level, of which Lachnospiraceae (including *Blautia* and *Coprococcus*), Ruminococcaceae (including *Oscillospira*), and Veillonellaceae (including *Veillonella dispar*) account for 63% of the selected OTUs, accompanied by Fusobacteriaceae (13%), Rikenellaceae (9%), Bacteroidaceae (9%), Enterobacteriaceae (6%), and Methylobacteriaceae (4%). For task II, *Gardnerella vaginalis* and *Lactobacillus vaginalis* account for 67% of the selected species, followed by *Gemella asaccharolytica* (13%), *Prevotella timonensis* (13%), and *P. amnii* (7%). For task III, a set of five genera (*Anaerococcus*, *Aerococcus*, *Corynebacterium*, *Lactobacillus*, and *Blautia*) account for 63% of the selected species, with special emphasis on *Anaerococcus hydrogenalis*, *Lactobacillus crispatus*, and *Blautia luti*. The subset identified for task IV was dominated by *Megasphaera micronuciformis* (48%), *Oribacterium sinus* (20%), *Lactobacillus salivarius* (13%), *Anaeroglobus geminatus* (11%), and *Fusobacterium periodonticum* (8%). Further analysis of the feature subsets selected by the best performing data inputs indicates a strong consistency for tasks II and IV (Figure 6), in which the same OTUs were consistently selected across the 25 runs. For tasks I and III, certain OTUs were selected almost consistently, but a larger proportion of the OTUs was selected by a few isolated GA runs only. This is especially true for task III, which had the lowest baseline score and, therefore, is considered to be the hardest classification task analyzed as part of this study.

## 4. Discussion

Microbiome datasets are often sparse and highly dimensional, meaning that not only do they present a significant amount of zeroes, since most microorganisms are not identified in all samples, but also, the number of samples is exceeded by the number of OTUs, or other components, such as genes and functional pathways, by an order of magnitude. These intrinsic characteristics are well-known challenges for the machine-learning field and can greatly affect the outcome of ML models. Herein, we present BiGAMi, a new feature selection method for microbiome data using genetic algorithms to tackle the dimensionality burden by reducing the number of OTUs used in ML classification tasks while retaining a high classification score. We also compare its results with the sequential forward selection (SFS) method provided by the *MLxtend* library, the SelectKBest method provided by *Scikit-learn*, GARS, and the baseline results for each study provided by Ref [29].

### 4.1. BiGAMi Drastically Reduced Microbiome Features for Classification Tasks

BiGAMi significantly reduced the number of microbiome features in all four tasks, identifying subsets of informative features hundreds of times smaller than the original dataset. The reference database used to map the OTUs had a small role in both the number of features in each subset and in the final classification score. The only task in which the database had a significant impact was task I, in which the Kostic colorectal cancer GG97 dataset led to better predictive performances than the RefSeq-mapped set, however, with a larger subset of features. The other tasks (II to IV) had AUC score differences that ranged from 0.01 to 0.04.

Differences in the input data (REL or CLR) showed a minor impact on the BiGAMi performance; however, CLR-transformed data resulted in a 2.6–2.8× lower number of selected OTUs than the relative abundance data. The use of CLR data has many advantages; for instance, to overcome differences in sequencing library sizes in Metabarcoding studies, the data have to be grouped in fractions (frequencies) to be compared between samples. Due to this fact, Metabarcoding data are strictly compositional, since they reside in a simplex rather than the Euclidean space [37] due to the sum constraint (frequencies of a sample sum to 1) and thus should be investigated using approaches developed by the compositional data analysis (CoDA) discipline.

Researchers have proposed data transformation approaches using ratios to remove the unit-sum constraint of compositional data and project it into the Euclidean space, such as the centered log-ratio transformation (CLR), additive log-ratio transformation (ALR), and isometric log-ratio transformation (ILR), of which CLR is most often used in multivariate data analysis [38,39]. Tools that consider the compositional nature of microbiome datasets have been published recently [40,41], and the microbiome field can greatly benefit from models using this type of data [42], or it can help circumvent the difficulties of dealing with zero values [43]. In addition, the combination of ML and CoDA has been successfully applied in a recent study to identify sources of potentially toxic elements in the soil of a mining city, in the field of geology [44].

### 4.2. BiGAMi Outperforms Other Feature Selection Methods

In this study, we compared the BiGAMi classification performance, as well as the size of the feature subsets leading to these performance metrics, against the results achieved by classical feature selection methods, such as sequential forward selection (SFS) and SelectKBest, and GARS, a different GA-based feature selection framework. The results of all experiments underline the value of BiGAMi’s bi-objective fitness function. In 29 out of 42 experiments, BiGAMi achieved a superior performance, either by increasing the classification metrics or by using a smaller feature subset of features. In six experiments, BiGAMi led to a higher performance score using a marginally larger feature subset than SFS or SelectKBest, and in seven experiments, it displayed a marginally lower performance score but still used fewer microbial features (Figure 5).

BiGAMi achieved a score better than or equal to the SFS method in 12 out of 14 experiments, with a reduction of up to 68% OTUs (Table 3). Only in two input data was the SFS performance marginally higher than BiGAMi’s performance metric at the cost of a larger OTU subset. Performance and OTU selection results for classification tasks, such as (II) Ravel vaginal Nugent category, where even the baseline result achieved a metric of 0.99, only differed marginally between SFS and BiGAMi. Due to its greedy mode of operation, in which each iteration adds the single feature with the largest gain in classification metric to the selected feature subset until a stop criterion is met, SFS lacks the capability of modeling and evaluating the complex feature interactions inherent to microbial datasets. Both algorithms were able to identify a limited number of OTUs needed to reliably classify samples into the correct categories; however, while both SFS and BiGAMi achieved the average classification performance per data input with comparable 99% confidence intervals (Figure 3), BiGAMi identified its 25 best performing OTU subsets more consistently around the average number of OTUs selected per data input (Figure 4). The SelectKBest method for selecting a fixed (user-configurable) number of relevant features relies solely on statistical dataset evaluations. Hence, like SFS, this methodology can also be blind to complex feature interactions. In comparison to SelectKBest, BiGAMi achieved, in 9 out of 14 experiments, a higher or equal classification score with up to 59% fewer OTUs (Table 3). In the remaining five experiments, SelectKBest marginally outperformed BiGAMi by 0.01–0.02 AUC classification metric at the cost of selecting substantially larger feature sets of 1.2× to 2.4× of the feature subset sizes selected by BiGAMi. Using an SGDClassifier without any form of OTU selection expectedly resulted in less performant classification results than those achieved with BiGAMi, SFS, or SelectKBest.

As another GA-based feature selection methodology, GARS was expected to be capable of modeling complex microbial feature interactions throughout the life cycle of the search process. Interestingly, with the exception of a single experiment (Task IV, RefSeq with CLR data transformation), GARS did not even reach the baseline classification results. On average, GARS selected feature subsets 5× the size of the feature subsets selected by BiGAMi, while mostly achieving significantly worse classification results. The reference implementation of GARS included running distinct GA searches per training data fold and thus resulting in overfitted fold-specific distinct feature subsets, which were mostly inconsistent with each other. Only on rare occasions, single OTUs were selected into each of the k-fold-specific feature subsets. For the fitness evaluation of GA individuals, GARS uses a random forest classifier, leading to longer runtimes than leveraging simpler linear models, as BiGAMi does. GARS results for relative abundance input data were often significantly worse than GARS performance on CLR transformed data, indicating a potentially hidden preference on the data representation. Lastly, GARS accepts the feature subset sizes of interest as input parameters, e.g., 5 to 20. GARS then runs separate feature selection searches per size (and fold) and tries to fully exploit a given subset size. In contrast, we designed BiGAMi to flexibly explore the number of active features in the GA population chromosomes led by the nature of a Darwinian search, resulting in largely reduced runtimes of just several minutes (BiGAMi) compared to multiple hours (GARS). Ultimately, BiGAMi showed increased performance in both classification score and the selection of a smaller subset of features, or at least one of both, thus highlighting its ability to reduce the high dimensionality of microbiome datasets.

### 4.3. BiGAMi Selects Features with Relevant Microbiological Role

The taxonomic information of the OTU subsets identified by BiGAMi suggested that BiGAMi extracts microbiologically relevant OTUs per classification task for well-known diseases, making this method suitable for reliable identification of relevant microbes for novel diseases or phenotypes in non-disease tasks.

In Task I subsets, several selected microbial families are well-known biomarkers for the detection of colorectal cancer [45,46,47,48,49]. Zhong et al., 2020, describe the relation of *Collinsella aerofaciens* and *Bacteroides*, among others, to the development of colorectal cancer. Gao et al., 2017, discovered that *Blautia* were significantly reduced in cancer patients, while *Bacteroides fragilis* and *Fusobacterium nucleatum* were enriched. Flemer et al., 2016, documented that cancer patients display an increased abundance of *Ruminococcus*.

For Task II subsets, we identified multiple recent publications that confirm that the selected OTUs, including *Gardnerella vaginalis* and *Lactobacilli*, are related to the development of bacterial vaginosis, which itself is diagnosed by a high vaginal Nugent score [50,51,52]. For Task III, it was already documented that Caucasian females have a vaginal microbiome dominated by *Lactobacillus crispatus,* among others, whereas women of African heritage show higher abundances of *Anaerococcus* and *Atopobium* [53]. Women of European ancestry, when diagnosed with bacterial vaginosis, were more likely to be colonized by *Corynebacterium*.

Lastly, for Task IV, the genus *Megasphaera*, among others, shows higher abundance counts in cirrhosis duodenum, while *Lachnospiraceae* show decreased abundances in a study with the salivary microbiome of cirrhotic patients [54]. The protective effect of *Lactobacillus salivarius* on liver injuries was already documented [55], as well as the fact that *Fusobacterium periodonticum* is enriched in cirrhosis patients [56].

The results presented here show that a bi-objective genetic algorithm fitness function helps in building and training well-performing host-state classifiers using a minimized subset of OTUs. Such models presented an improved predictive performance when compared to the baseline models [29] and also exceeded or matched the performance of the other algorithms on almost all data inputs while retaining smaller subsets of features. At the same time, BiGAMi achieves these classification performance results by drastically reducing the number of predictive OTUs compared to other algorithms. The use of a fitness function that merges the actual classification performance and the chromosome size of an individual into a single metric is essential for guiding the GA search toward finding high-performance OTU subsets and proved to work efficiently on microbiome datasets.

This study only evaluated the effectiveness of the GA-based OTU selection on classification problems. It is expected that additional regularization operations are required to trade off the regression metric and the number of selected OTUs. General GA search parameters, such as the number of generations and population size, were selected in a way that limited computation capacity, leading to superior results. It remains for future research to define parameter guidelines that produce similar results with reduced computational cost.

## 5. Conclusions

This study demonstrated the successful application of a genetic algorithm with a bi-objective fitness function to select the most predictive combination of OTUs from microbiome datasets to classify host phenotypes. It was shown that such a GA evolutionary search for the most predictive feature (OTU) subset improves classification performance for all classification problems. Where classifiers without a feature selection already achieved almost perfect results, our proposed BiGAMi method performed “on par”. Furthermore, BiGAMi achieved its results by selecting significantly fewer OTUs than other methods we compared our results with (up to 68% fewer OTUs than sequential forward selection, up to 59% fewer OTUs than SelectKBest, and up to 93% fewer OTUs than GARS).

BiGAMi selected, on average, 1.02% of the original number of OTUs across the 14 experiments, reducing the feature space by two orders of magnitude. Compared to methods relying on the adoption of deep learning and variational autoencoders, this feature space reduction helps simpler classifiers to find patterns in the data more easily and improves the interpretability of the classification results. This is a desirable capability, especially for machine-learning models used for medical diagnoses.

## Figures and Tables

**Figure 1 mps-05-00042-f001:**
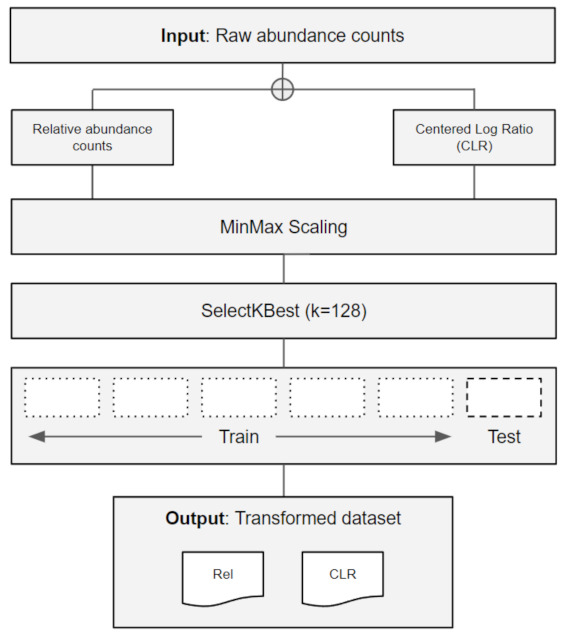
Data pre-processing flow from raw abundance counts input data to the transformed datasets used in classification tasks.

**Figure 2 mps-05-00042-f002:**
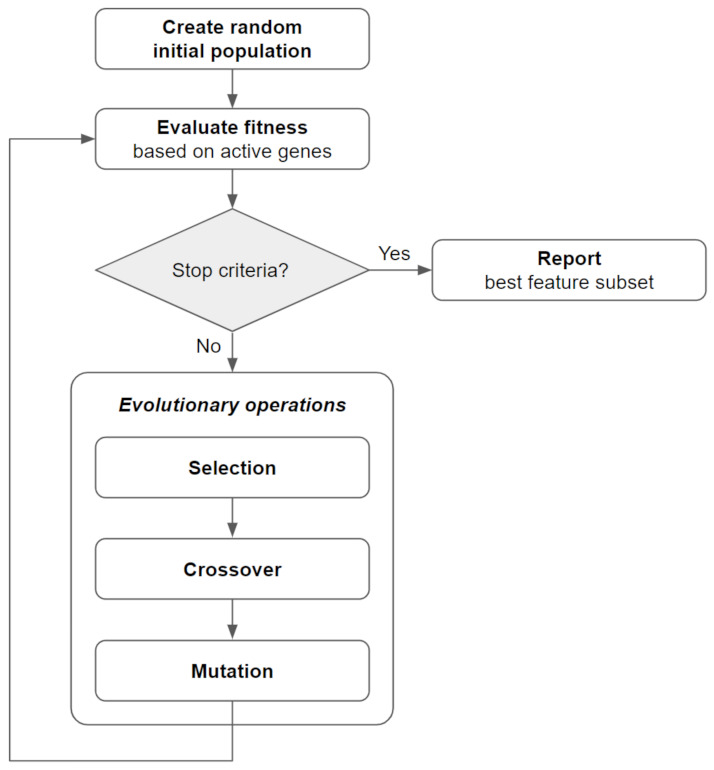
General process flow of a genetic algorithm.

**Figure 3 mps-05-00042-f003:**
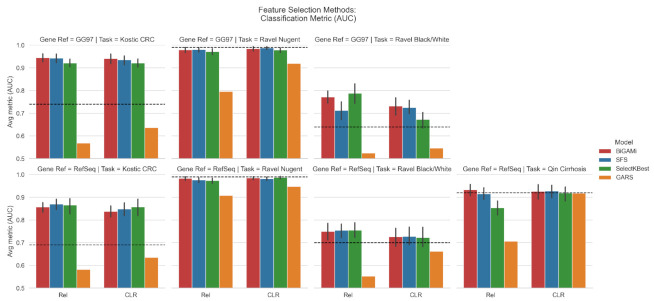
Average classification performance (including 99% confidence interval) for each data input achieved by the BiGAMi, SFS, and GARS methods. Confidence intervals were calculated by the *Seaborn* plotting library using the bootstrap resampling technique with a mean estimator. The dashed horizontal lines represent the base performances as documented by Ref [29]. (**Top**): Performance results for GG97-based data input. (**Bottom**): Performance results for RefSeq-based data input.

**Figure 4 mps-05-00042-f004:**
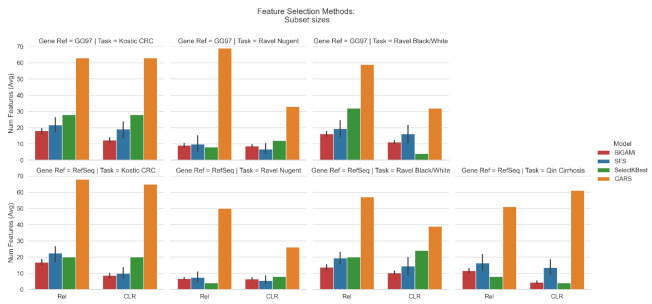
Average number of selected OTUs (including 99% confidence interval) for each data input achieved by the BiGAMi, SFS, and GARS methods. Confidence intervals were calculated by the *Seaborn* plotting library using the bootstrap resampling technique with a mean estimator. (**Top**): Selected OTUs for GG97-based data input. (**Bottom**): Selected OTUs for RefSeq-based data input.

**Figure 5 mps-05-00042-f005:**
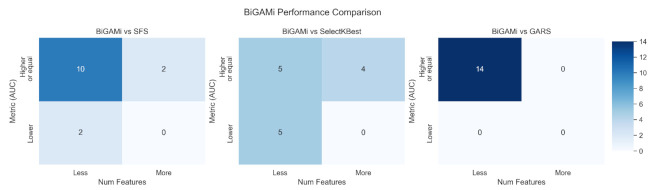
Performance comparison overview between BiGAMi, SFS, SelectKBest, and GARS. The upper left cells are preferred, indicating BiGAMi achieves a higher classification metric using a smaller feature subset.

**Figure 6 mps-05-00042-f006:**
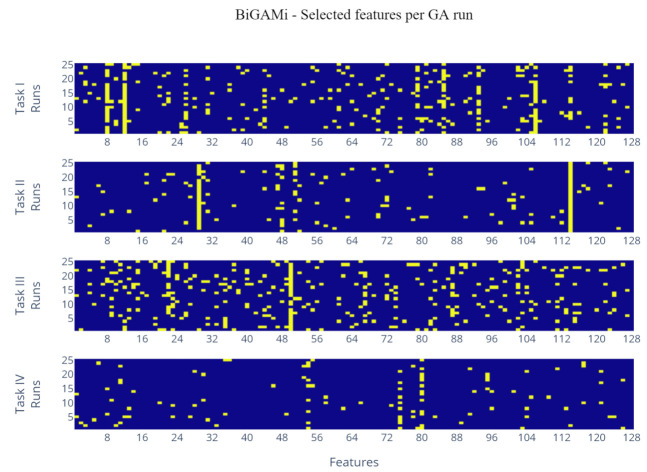
Feature selection heatmap per best performance data input. *Runs* represent the 25 BiGAMi runs per data input. *Features* represent the 128 features selected by the SelectKBest operation for each data input. Yellow marks indicate that a feature was part of the best performing GA individual feature subset. Vertical structures of yellow marks signal that a certain feature was consistently identified as being highly relevant across the 25 BiGAMi runs. Task I: Kostic colorectal cancer healthy/tumor GG97 CLR, Task II: Ravel vaginal Nugent category RefSeq CLR, Task III: Ravel vaginal black/white RefSeq Rel, Task IV: Qin cirrhosis RefSeq CLR.

**Table 1 mps-05-00042-t001:** Description of datasets used in this study.

	Task I	Task II	Task III	Task IV
Dataset	Kostic [30]	Ravel [31]	Ravel [31]	Qin [6]
Year	2012	2011	2011	2014
Description	Healthy vs. Tumor Colon Biopsy Tissues	Low vs. High Vaginal Nugent Score	Black vs. White phenotype classification	Cirrhosis vs. healthy
Topic area	Colorectal Cancer	Vaginal	Vaginal	Cirrhosis
Classification targets	Healthy, Tumor	Low, High	Black, White	Cirrhosis, Healthy
Number of samples	190	342	200	130
Number of subjects	95	342	200	130
Number of OTUs GG97	3228	1093	1093	n/a
Number of OTUs RefSeq	908	586	586	8483

**Table 2 mps-05-00042-t002:** GA parameters.

Parameter	Description	Value
n_searches	Number of individual GA runs	25
pop_size	GA population size	250
max_iter	Number of GA iterations/generations	10
bestN	Elitism concept	1
crossover	Crossover strategy	1p (One-point)
CXPB	Crossover probability	0.8
MUPB	Mutation probability	0.8
init	GA individual initialization strategy	zero
init_ind_length	Average number of enabled GA individual chromosomes	10
select	GA crossover selection strategy	Tournament (size = 3)
mutate	GA mutation operation	mutFlipOne (custom)

**Table 3 mps-05-00042-t003:** Performance results per classification task. Per input data, the average AUC score and the average number of OTUs are provided. (I) Kostic colorectal cancer healthy/tumor GG97, (II) Ravel vaginal Nugent category, (III) Ravel vaginal black/white, (IV) Qin cirrhosis RefSeq. Rows with an * symbol were selected for taxonomy analysis of selected feature subsets.

Task	Database	Total OTUs	Baseline AUC	Input Data	SGD AUC	SFS AUC/OTUs	SelectKBest AUC/OTUs	GARS AUC/OTUs	BiGAMi AUC/OTUs (Ours)
(I)	GG97	3228	0.74	Rel	0.85	0.94/21.6	0.92/28	0.57/63	0.95/18.1
				CLR	0.9	0.94/19.0	0.92/28	0.64/63	0.94/12.3
	RefSeq	908	0.69	Rel	0.8	0.87/22.4	0.87/20	0.58/68	0.86/16.8
				CLR	0.85	0.85/10.0	0.86/20	0.64/65	0.84/8.2
(II)	GG97	1093	0.99	Rel	0.96	0.98/9.8	0.97/8	0.80/69	0.98/9.2
				CLR	0.97	0.99/6.6	0.98/12	0.92/33	0.99/8.6
	RefSeq	586	0.99	Rel	0.96	0.98/7.4	0.97/4	0.91/50	0.98/6.6
				CLR	0.96	0.98/5.5	0.99/8	0.95/26	0.98/6.5
(III)	GG97	1093	0.64	Rel	0.65	0.71/19.4	0.79/32	0.52/59	0.77/16.2
				CLR	0.63	0.73/16.2	0.67/4	0.55/32	0.73/11.1
	RefSeq	586	0.7	Rel	0.6	0.75/19.4	0.76/20	0.55/57	0.75/13.6
				CLR	0.61	0.73/14.4	0.72/24	0.62/39	0.73/10.2
(IV)	RefSeq	8483	0.92	Rel	0.82	0.92/16.4	0.85/8	0.71/51	0.93/11.6
				CLR	0.83	0.93/13.5	0.92/4	0.92/61	0.93/4.3

## Data Availability

The datasets used for this study are accessible via the following links: https://github.com/knights-lab/MLRepo/tree/master/datasets/kostic; https://github.com/knights-lab/MLRepo/tree/master/datasets/qin2014 (accessed on 15 March 2021). The code generated for the BiGAMi framework is available on GitHub: https://github.com/mikeleske/BiGAMi (accessed on 10 April 2022).

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
