# Peer review of "BiGAMi: Bi-Objective Genetic Algorithm Fitness Function for Feature Selection on Microbiome Datasets"

_mps, 2022, doi:10.3390/mps5030042_

Round 1

Reviewer 1 Report

The authors proposed a bi-objective genetic algorithm fitness function, called BiGAMi, for feature selection in microbial datasets. It is meaningful to research this topic. Experimental results demonstrate the proposed method is useful. But the following problems should be declared: 

1. It is suggested to add more description of the method in lines 55-61.

2. Figure 4 and Figure 5 can be represented by Table 3, and I suggest adding descriptions of Figure 4 and Figure 5 that Table 3 cannot observe.

3. I can't read the description of lines 192-209 correctly because Figure 3 is not high enough resolution.

4. Is the code available on GitHub (https://github.com/mikeleske/BiGAMi)?

Author Response

It is suggested to add more description of the method in lines 55-61. 

Dear Reviewer, we thank you for highlighting this. The paper went through rigorous re-writing, and we believe that the description is satisfactory now. 

Figure 4 and Figure 5 can be represented by Table 3, and I suggest adding descriptions of Figure 4 and Figure 5 that Table 3 cannot observe. 

Dear reviewer, we improved the Figure 4 and 5 descriptions.  

I cannot read the description of lines 192-209 correctly because Figure 3 is not high enough resolution. 

Dear reviewer, we are deeply sorry for that. We worked on the figures and provided a higher resolution version of them. 

Is the code available on GitHub (https://github.com/mikeleske/BiGAMi)? 

Dear reviewer, yes, the source code is available at this address. 

Reviewer 2 Report

The authors proposed a bi-objective genetic algorithm fitness function for feature selection in microbiome datasets. The proposed fitness function is used for training phenotype classifiers.

Unfortunately, the paper contains the following problems.

  • The paper language needs polishing since it contains syntax errors.

  • The literature review part at the end of the “Introduction” section needs significant expansion. The authors must include more papers regarding existing feature selection methods and feature selection methods focused on microbiome features. The total referenced articles must be at least 30, followed by a brief explanation of each work.

  • In the expanded literature review part of the “Introduction” section, the authors must explain their proposed method’s (BiGAMi) advantage(s) over the recently added ones.

  • At the end of the “Introduction” section, a small paragraph explaining the paper’s structure must be included.

  • The characteristics of the microbiome datasets used in the experimental part (e.g., sizes, number of features, etc.) must be explained. Also, consider adding a table summarizing these characteristics.

  • In section 2.2, a table summarizing the parameters used in the genetic algorithm-based feature selection method must be added.

  • A section describing the structure of a generic genetic algorithm followed by pseudocode, a diagram, and a thorough explanation must be included. This section must be placed before section 2.2.

  • The experimental part compares BiGAMi with only one method. Testing BiGAMi with only one other approach cannot significantly indicate that it is better than the existing ones. The authors must provide a comparison with at least five methods.

  • In section 3, a table summarizing the experiments’ parameters must be added.

  • The experimental part does not mention which classifier was used for the classification tasks.

  • The authors do not explain how the confidence intervals shown in Figures 4 and 5 were calculated.

  • In Figure 4, which describes the classification performance of BiGAMi with another existing method, the 99% confidence intervals have partial or complete overlaps in all cases. This finding cannot allow us to conclude the significance of BiGAMi. A strong indication that the results from the proposed method are significantly better than the compared one would be the absence of overlaps in the confidence intervals.

Author Response

The paper language needs polishing since it contains syntax errors. 

Dear Reviewer, we thank you for highlighting this. The paper went through rigorous re-writing, and we believe that the language is now suitable for publication. 

The literature review part at the end of the “Introduction” section needs significant expansion. The authors must include more papers regarding existing feature selection methods and feature selection methods focused on microbiome features. The total referenced articles must be at least 30, followed by a brief explanation of each work. 

Dear Reviewer, the introduction section was significantly expanded with the inclusion of pertinent literature. 

In the expanded literature review part of the “Introduction” section, the authors must explain their proposed method’s (BiGAMi) advantage(s) over the recently added ones. 

Dear Reviewer, in the introduction we now described the bi-objective function and why we choose this structure and why is it superior for the problem of dimensionality. We compared the method with other feature selection methods over the manuscript and decisively showed that BiGAMi outperforms other feature selection methos. 

At the end of the “Introduction” section, a small paragraph explaining the paper’s structure must be included. 

Dear Reviewer, we included a paragraph at the end of the introduction describing the datasets and experiments. 

The characteristics of the microbiome datasets used in the experimental part (e.g., sizes, number of features, etc.) must be explained. Also, consider adding a table summarizing these characteristics. 

Dear Reviewer, we included a text describing the datasets, as well as a table summarizing their characteristics (table 1). 

In section 2.2, a table summarizing the parameters used in the genetic algorithm-based feature selection method must be added. 

Dear Reviewer, we included a table summarizing the used parameters (table 2) 

A section describing the structure of a generic genetic algorithm followed by pseudocode, a diagram, and a thorough explanation must be included. This section must be placed before section 2.2. 

Dear reviewer, we included the thorough explanation of GAs in the manuscript, thanks for pointing it out. 

The experimental part compares BiGAMi with only one method. Testing BiGAMi with only one other approach cannot significantly indicate that it is better than the existing ones. The authors must provide a comparison with at least five methods. 

Dear Reviewer, the number of feature selection algorithms available for Microbiome data is scarce, but we compared BiGAMi with SFS, SelectKbest and another Genetic Algorithm based feature selection algorithm (GARs). 

In section 3, a table summarizing the experiments’ parameters must be added. 

Dear reviewer, we included a table 3 summarizing the experiments. 

The experimental part does not mention which classifier was used for the classification tasks. 

Dear reviewer, we included the classifier used by the end of section 2, it was the scikit-learn SGDClassifier. 

The authors do not explain how the confidence intervals shown in Figures 4 and 5 were calculated. 

Dear reviewer, the confidence intervals were not generated by the classification model but from the visualisation function “catplot”. They were standard deviations from the 25 individual runs of each method.

In Figure 4, which describes the classification performance of BiGAMi with another existing method, the 99% confidence intervals have partial or complete overlaps in all cases. This finding cannot allow us to conclude the significance of BiGAMi. A strong indication that the results from the proposed method are significantly better than the compared one would be the absence of overlaps in the confidence intervals. 

Dear Reviewer, indeed, the classification performance of BiGAMi was comparable to SFS in most experiments. However, although having similar classification scores, BiGAMi was a superior feature selection in all but 2 cases, since it selected a smaller subset of features when compared to SFS.  

Reviewer 3 Report

 I think the application of the Genetic Algorithm in this study is well presented. The presentation is good, and also the evaluation experiments are comprehensive. The authors also provided their codes in a public platform to make it easier to the reader to reproduce the paper. The paper is completely acceptable for publication with only some minor revision, as follows:

  • It will be better if you can clarify some text in the figures, some of them can not be read.
  •  It will be better if you add recent applications of some optimization algorithms that are used as FS algorithms, such as: Boosted ANFIS model using augmented marine predator algorithm with mutation operators for wind power forecasting; Novel Improved Salp Swarm Algorithm: An Application for Feature Selection; Modified marine predators algorithm for feature selection: case study metabolomics; Novel Improved Salp Swarm Algorithm: An Application for Feature Selection; A Review of the Modification Strategies of the Nature Inspired Algorithms for Feature Selection Problem; Advanced feature extraction and selection approach using deep learning and Aquila optimizer for IoT intrusion detection system;
  • A light proofreading is needed.

Author Response

It will be better if you can clarify some text in the figures, some of them cannot be read. 

Dear reviewer, we are deeply sorry for that. We worked on the figures and provided a higher resolution version of them. 

It will be better if you add recent applications of some optimization algorithms that are used as FS algorithms, such as: Boosted ANFIS model using augmented marine predator algorithm with mutation operators for wind power forecasting; Novel Improved Salp Swarm Algorithm: An Application for Feature Selection; Modified marine predators algorithm for feature selection: case study metabolomics; Novel Improved Salp Swarm Algorithm: An Application for Feature Selection; A Review of the Modification Strategies of the Nature Inspired Algorithms for Feature Selection Problem; Advanced feature extraction and selection approach using deep learning and Aquila optimizer for IoT intrusion detection system; 

Dear Reviewer, the number of feature selection algorithms available for Microbiome data is scarce, but we compared BiGAMi with SFS, SelectKbest and another Genetic Algorithm based feature selection algorithm (GARs). Due to the short deadline, we could not run all the algorithms you proposed but we certainly will in future versions of this algorithm, as we are planning to expand it to regression models as well 

A light proofreading is needed. 

Dear reviewer, the paper went through rigorous re-writing, English checking, and proofreading. 

Round 2

Reviewer 2 Report

The authors addressed most of my comments.